# Soybean Thrips (Thysanoptera: Thripidae) Harbor Highly Diverse Populations of Arthropod, Fungal and Plant Viruses

**DOI:** 10.3390/v12121376

**Published:** 2020-12-01

**Authors:** Thanuja Thekke-Veetil, Doris Lagos-Kutz, Nancy K. McCoppin, Glen L. Hartman, Hye-Kyoung Ju, Hyoun-Sub Lim, Leslie. L. Domier

**Affiliations:** 1Department of Crop Sciences, University of Illinois, Urbana, IL 61801, USA; tthekke2@illinois.edu; 2Soybean/Maize Germplasm, Pathology, and Genetics Research Unit, United States Department of Agriculture-Agricultural Research Service, Urbana, IL 61801, USA; doris.lagos-kutz@usda.gov (D.L.-K.); nancy.mccoppin@usda.gov (N.K.M.); glen.hartman@usda.gov (G.L.H.); 3Department of Applied Biology, College of Agriculture and Life Sciences, Chungnam National University, Daejeon 300-010, Korea; jhk91113@gmail.com (H.-K.J.); hyounlim@cnu.ac.kr (H.-S.L.)

**Keywords:** viral metatranscriptomics, vector-enabled transcriptomics, soybean thrips, *Neohydatothrips variabilis*

## Abstract

Soybean thrips (*Neohydatothrips variabilis*) are one of the most efficient vectors of soybean vein necrosis virus, which can cause severe necrotic symptoms in sensitive soybean plants. To determine which other viruses are associated with soybean thrips, the metatranscriptome of soybean thrips, collected by the Midwest Suction Trap Network during 2018, was analyzed. Contigs assembled from the data revealed a remarkable diversity of virus-like sequences. Of the 181 virus-like sequences identified, 155 were novel and associated primarily with taxa of arthropod-infecting viruses, but sequences similar to plant and fungus-infecting viruses were also identified. The novel viruses were predicted to have positive-sense RNA, negative-stranded RNA, double-stranded RNA, and single-stranded DNA genomes. The assembled sequences included 100 contigs that represented at least 95% coverage of a virus genome or genome segment. Sequences represented 12 previously described arthropod viruses including eight viruses reported from Hubei Province in China, and 12 plant virus sequences of which six have been previously described. The presence of diverse populations of plant viruses within soybean thrips suggests they feed on and acquire viruses from multiple host plant species that could be transmitted to soybean. Assessment of the virome of soybean thrips provides, for the first time, information on the diversity of viruses present in thrips.

## 1. Introduction

Thrips (order, *Thysanoptera*) are economically important agricultural pests. In addition to the damage caused by feeding, thrips transmit some of the most damaging plant viruses including members of the genera *Carmovirus*, *Ilarvirus*, *Machlomovirus, Orthotospovirus*, and *Sobemovirus* [1]. Viruses in the genus *Orthotospovirus* (family, *Tospoviridae*; e.g., iris yellow spot virus and tomato spotted wilt virus [TSWV]) cause significant economic losses in multiple agricultural crops worldwide [2,3]. Soybean vein necrosis orthotospovirus (SVNV) is an emerging virus in soybean that has widespread occurrence in the United States (US) and Canada [4]. SVNV causes vein clearing, leaf chlorosis, and leaf necrosis in soybean [5] and was reported to reduce seed quality and oil content [6]. SVNV is exclusively transmitted by thrips in a persistent and propagative manner, primarily by soybean thrips (*Neohydatothrips variabilis*, Beach) [7] and to a lesser extent by Eastern flower thrips (*Frankliniella tritici,* Fitch) and tobacco thrips (*F. fusca,* Hinds) [8].

Soybean thrips were originally reported on smartweed and cucumber [9] and have been found on cotton [10], lima bean [11], soybean [12], and tomato [13] in the U.S. SVNV is the only virus reported to be transmitted by soybean thrips and there is a lack of information regarding the vector potential of soybean thrips for other plant viruses. Additionally, no information is available about viruses that infect thrips and those that are circulated by thrips without regard to their transmission potential. 

Virus metagenomics or metatranscriptomics has been used to discover and characterize diverse viruses present in selected organisms by high-throughput sequencing (HTS). With the increasing use of HTS, the information gathered on the diversity of viral communities in various organisms/environments are rapidly expanding. Vector-enabled metagenomics or metatranscriptomics (VEM), where viruses are sequenced directly from insect vectors, can characterize virus diversity in arthropods and indirectly, the hosts on which they feed. The VEM approach has been successfully used to detect previously described viruses and discover novel viruses from insect vectors of plant viruses such as aphids [14,15], leaf hoppers [16,17], psyllids [18], and whiteflies [19,20,21]. 

In this communication, we present information gathered on a broad range of insect, plant, and fungal viruses detected in soybean thrips through metatranscriptome analysis. Our study provides unique insights into the virus diversity in thrips for the first time, particularly soybean thrips, and suggests that VEM when paired with systematic vector sampling could be used as a molecular surveillance system capable of recognizing novel insect viruses and emerging plant viruses of potential agricultural importance. 

## 2. Materials and Methods

### 2.1. Collection and Processing of Soybean Thrips Samples

Soybean thrips were captured in 2018 by the Suction Trap Network [22], which is located in Illinois (Freeport, Monmouth, Morris and Urbana-Champaign), Iowa (Ames, Kanawha, Nashua and Sutherland), Kansas (Manhattan), Louisiana (Chase), Michigan (Kellogg and Monroe), Minnesota (Lamberton, Morris and Rosemount), Missouri (Columbia), and Wisconsin (Arlington, Eau Claire and Hancock). Thrips were manually selected from the suction traps, transferred to 95% ethanol, and stored at −20 °C. Soybean thrips were identified based on the morphological characteristics included in Hoddle et al. [23]. A total of 15,275 soybean thrips were collected and processed from Illinois (5691), Indiana (1583), Iowa (4401), Kansas (523), Louisiana (418), Michigan (357), Minnesota (1729), Missouri (49), and Wisconsin (254). The thrips were pooled, and total RNA was extracted from the pooled samples using the RNeasy Mini Kit (Qiagen, Valentia, CA, USA), treated with Turbo DNase (ThermoFisher, Waltham, MA, USA), and depleted of ribosomal RNA (rRNA) with the Illumina Ribo-Zero Plus rRNA Depletion Kit (Illumina, San Diego, CA, USA). Initially, a single-end 100-nucleotide (nt) sequencing library was prepared from the rRNA-depleted sample using the Illumina ScriptSeq RNA-Seq Library Preparation Kit and sequenced on an Illumina NovaSeq 6000 at the Roy J. Carver Biotechnology Center at the University of Illinois. Due to the complexity and large number of partial virus sequences in the first sequencing run, a second paired-end, 250-nt library was prepared and sequenced.

### 2.2. Bioinformatic Analyses

Reads from each library were assembled de novo with Trinity [24] and SPAdes [25]. Assembled contigs were compared to a customized database that included predicted reference invertebrate and virus amino acid (aa) sequences (NCBI GenBank Release 236) using USEARCH [26]. Contigs with significant similarity to virus sequences were compared to the NCBI nonredundant protein database using BLASTX [27]. The number of reads aligning to and depths of coverage for selected sequences were calculated using Bowtie2 [28] and SAMtools [29]. For phylogenetic analyses, RNA-dependent RNA polymerase (RdRp) sequences were aligned using MUSCLE [30], and maximum likelihood trees were constructed using MEGA 7 [31]. As RdRp domain containing sequences were not available for all sequences, the capsid protein (CP [VP2]) sequences were used for additional analysis of *Permutotetraviridae* members and glycoprotein (GP) sequences were used for *Chuviridae* members.

### 2.3. Nucleotide Sequence Accession Numbers

Sequence data are available at the National Center for Biotechnology Information Short Read Archive (SRA) under accession number PRJNA614937. GenBank accession numbers for individual virus-like sequences are given in Appendix A.

## 3. Results

### 3.1. Virus-Like Sequences Associated with Soybean Thrips

The first sequencing library produced 2.3 × 10^8^ 100-nt reads and the second library produced 1.1 × 10^9^ 250-nt paired-end reads. As a reference genome or transcriptome was not available for soybean thrips, it was not possible to align the sequence reads or contigs to a cognate sequence. However, 70.2% of the contigs assembled from the second library with lengths of at least 1000 nt aligned to the reference proteome of *F. occidentalis*, the most closely related species for which a genome sequence is available. The next three species with significant numbers of contigs aligning were *Lucilia cuprina* Wiedemann (Australian sheep blowfly; 15.3% contigs aligning), *Contarinia nasturtii* Kieffer (swede midge; 6.7% contigs aligning), and *Nasonia vitripennis* Walker (parasitoid wasp; 5.0% contigs aligning). Only about 0.1% of the contigs aligned to bacterial endosymbionts of invertebrates.

From the two sequencing libraries, approximately 0.85% were predicted to be of viral origin. The SPAdes assembler sometimes produced longer contigs than the Trinity assembler, but many times, the longer contigs had assembly errors. Among the assembled sequences, 100 contigs represented at least 95% coverage of the genomes or genome segments of the most closely related virus species (Appendix A). The mean depth of coverage of the assembled virus-like sequences ranged from 0.6 to 38,908-fold for the selected contigs. A total of 181 virus-like sequences were recovered from the transcriptome data. Among these sequences, 103 were related to positive-sense RNA viruses in the orders *Amarillovirales* (20), *Martellivirales* (13), *Ourlivirales* (3), *Permutotetraviridae* (4), *Picornavirales* (31), *Sobelivirales* (12), *Tolivirales* (13), *Tymovirales* (3), and *Wolframvirales* (4). Fifty-four sequences were related to negative-stranded RNA viruses in the orders *Articulavirales* (15), *Bunyavirales* (20), *Jingchuvirales* (9), *Mononegavirales* (5), and *Serpentovirales* (5). Twenty-two sequences were related to double-stranded RNA (dsRNA) viruses in the orders *Durnavirales* (14), *Ghabrivirales* (5), and *Reovirales* (3). Finally, two sequences were related to single-stranded (ss) DNA viruses in the order *Piccovirales*. The three most abundant viruses in the combined sample were soybean thrips iflavirus (STIV) 2 (2,350,342 reads), followed by Hubei arthropod virus (HAV) 1 (834,888 reads), and STIV8 (743,592 reads) (Appendix A).

### 3.2. Sequences Related to Single-Stranded Positive-Sense RNA Viruses

#### 3.2.1. Sequences Related to Members of the *Picornavirales*

The picorna-like virus sequences obtained from the soybean thrips transcriptome data resembled the genomes of members of the *Caliciviridae*, *Dicistroviridae*, and *Iflaviridae* families and unclassified picorna-like viruses. The assembled contigs included 13 sequences (MT195546-MT195548, MT224138-MT224142, MT240781, MT293153, MW023868, MW039357, and MW039359) with similarity to members of the *Iflaviridae*, which included a sequence that was 96.4% identical to HAV1 (YP_009336629.1) and sequences of 12 putative new iflaviruses (Appendix A). Members of the *Iflaviridae* infect arthropods and have monopartite positive-sense ssRNA genomes containing a single open reading frame (ORF) encoding a polyprotein [32] (Figure 1). For nine of the 12 new virus sequences, HTS provided greater than 90% genome coverage (Appendix A) of which STIV1-6 and STIV8 contained presumably full-length polyprotein sequences (329–369 kDa). The remaining sequences contained partial polyproteins. In phylogenetic analysis, these sequences were grouped with recognized members of the *Iflaviridae* (Figure 2).

Nine dicistrovirus-like sequences were assembled from the HTS data. The *Dicistroviridae* is a family of arthropod-infecting viruses whose genomes contain two non-overlapping ORFs [33]. The 5′-proximal ORF encodes the nonstructural proteins, and the 3′-proximal ORF encodes the structural proteins. Three contigs represented described viruses: aphid lethal paralysis virus (ALPV; MT240796), and Rhopalosiphum padi virus (RhPV; MW039360 and MW039361) (Appendix A). The remaining six sequences represented novel virus genomes. Two of the sequences had genome organizations typical of the *Dicistroviridae* and were tentatively named soybean thrips dicistrovirus (STDV) 1 (MT195550) and STDV2 (MT224137). The nearly complete genomes of STDV1 and STDV2 were predicted to encode 231 kDa and 218 kDa nonstructural polyproteins, respectively, and 96 kDa and 117 kDa structural polyproteins, respectively (Figure 1). Soybean thrips picorna-like virus (STPiLV) 3 (MT240798), STPiLV4 (MT240799), and STPiLV10 (MW023863) were partial genome sequences that contained only RdRp coding regions, and therefore their genome organizations could not be determined although they grouped with recognized members of the *Dicistroviridae* in phylogenetic analysis (Figure 3). Therefore, they were provisionally named with the more generic names “picorna-like virus”. STPiLV1 (MT240797) had a genome arrangement similar to members of the family *Marnaviridae* where the genome encodes a single polyprotein (303 kDa) with the nonstructural proteins at the N-terminus and the structural proteins at the C-terminus of the polyprotein (Figure 1). However, RdRp of STPiLV1 grouped phylogenetically with members of the *Dicistroviridae*, indicating the diverse nature of soybean thrips dicistroviruses (Figure 2). 

The predicted aa sequence of contig (MT293133) showed 95% identity to the polyprotein (77% sequence coverage) of Pernambuco virus (MK189088), an unclassified RNA virus identified in Brazil from ruddy turnstones (*Arenaria interpres*) [34]. The sequence contained a RdRp domain (pfam00680), a dsRNA binding motif (DSRM_SF; cd00048), and a calicivirus CP domain (pfam00915) (Figure 1). In phylogenetic analysis, this sequence grouped with unclassified RNA viruses discovered from fruit flies (Figure 2). The genomes of these viruses also encoded a single polyprotein with similar conserved domains and motifs as those of soybean thrips Pernambuco virus. The monophyletic grouping of these viruses suggests that Pernambuco virus is an arthropod virus. Although these viruses possess a calicivirus CP domain, they did not group with recognized members of the *Caliciviridae* (Figure 2).

One contig represented a nearly complete genome of a picorna-like bicistronic virus, tentatively named soybean thrips bicistronic virus 1 (STBV1; MT195549) that had two large ORFs (Figure 1). The ORFs were predicted to encode proteins with similarity to structural and nonstructural proteins of members of the *Picornavirales*. The genome organization resembled that of members of the *Dicipivirus* genus in the *Picornaviridae* where the 5’-proximal ORF encodes the structural polyprotein (91 kDa) and the 3’-proximal ORF encodes the nonstructural polyprotein (193 kDa). This arrangement of ORFs is the opposite of the *Dicistroviridae*. Recently, viruses with similar genome organizations have been reported from plants [35], bats [36], and arthropods [37,38,39]. In phylogenetic analysis, STBV1 grouped with unclassified bicistronic viruses that are presumed to infect arthropods (Figure 2). 

One contig (MT293126) was 95% identical to the genome sequence of Hubei picorna-like virus 55 (HPiLV55; NC_033093.1). Although HPiLV55 possesses a genome organization typical for members of the *Dicistroviridae*, its RdRp was phylogenetically distant from recognized *Dicistroviridae* members. HPiLV55 branched separately, but shared a monophyletic origin with a group of unclassified RNA viruses that included soybean thrips Pernambuco virus (Figure 2). Like these viruses, the HPiLV55 nonstructural polyprotein was also predicted to contain a dsRNA binding motif, but it differed from them in having a bicistronic genome and lacked the calicivirus CP domain in the structural polyprotein. Other partial genomes of picorna-like viruses in the assembled data (Appendix A) were not included in the phylogenetic analysis because they lacked RdRp conserved domains. 

#### 3.2.2. Sequences Related to Members of the *Sobelivirales*

Of the 12 assembled sequences that were related to members of the *Solemoviridae*, 11 were novel virus-like sequences of which 10 resembled the genomes of unclassified sobemo-like viruses reported from arthropods (Appendix A). Except for soybean thrips sobemo-like virus (STSLV) 7 (MW039353) and STSLV9 (MW039355), these sequences contained two overlapping ORFs (Figure 1) in which the second ORF was predicted to be translated by a −1 frameshift. ORF1 encoded a protein (p1, 56–83 kDa) of unknown function, while ORF2 was predicted to encode proteins of 35–61 kDa with homology to RdRps with pfam02123 or pfam00680 domains. In STSLV2 (MT293132), STSLV3 (MW023865), and STSLV6 (MW039352), the p1 protein was predicted to contain a trypsin-like peptidase domain. The partial genome of STSLV11 (MW023864) encoded a nonstructural polyprotein, P2ab, and a CP, which showed the highest aa sequence identity to orthologues in sesbania mosaic virus (62%; NP_066393.4) and velvet tobacco mottle virus (36%; YP_003896040.1), respectively, of the *Sobemovirus* genus. A nearly complete genome sequence of Lucerne transient streak virus (MT224146), a plant-infecting member of the *Sobemovirus* genus, was also assembled. LTSV has been reported from forage legumes in Canada, but not in the U.S. [40,41]. In phylogenetic analysis of RdRp aa sequences, STSLV1 to STSLV10 branched with unclassified sobemo-like viruses reported from arthropods, while STSLV11 and LTSV branched with plant-infecting *Sobemovirus* members (Figure 3). 

#### 3.2.3. Sequences Related to Members of the *Martellivirales*

Nine novel virus-like sequences with significant similarity to members of the order *Martellivirales* were obtained from soybean thrips metatranscriptome data. Two were tentatively named soybean thrips virga-like virus (STVLV) 1 (MT240784) and STVLV2 (MT240785). The 10,947-nt genome of STVLV1 was greater than 90% complete and contained five ORFs (Figure 1). The predicted aa sequence of ORF1 was most similar to the predicted polyprotein of Beult virus, an unclassified RNA virus (AWA82275.1) from *Drosophila suzukii* [42]. ORF1 encoded a 295 kDa nonstructural polyprotein that contained methyltransferase (MTR, pfam01660), Superfamily (SF) 1 RNA helicase (HEL, pfam01443), membrane-attack complex/perforin (MACPF; PTZ00482), and RdRp (pfam00978) domains. ORFs 2, 3, 4, and 5 were predicted to encode proteins of 16 kDa, 20 kDa, 59 kDa, and 17 kDa, respectively, of which the product of ORF3 showed 23% aa sequence identity with a hypothetical protein from Wuhan heteroptera virus 1 (YP_009342333.1). 

Two sequences, soybean thrips nege-like virus (STNLV) 1 (MT240782) and STNLV2 (MW039375 and MW039377), were similar to members of the proposed Negevirus taxon of insect-specific viruses from mosquitoes and phlebotomine sandflies that have 9–10 kb positive-sense ssRNA genomes [43,44]. The genome of STNLV1 (9369 nt) contained three ORFs (Figure 1). ORF1 was predicted to encode a 273 kDa nonstructural polyprotein containing two putative MTR domains: pfam01660, which is found in a wide range of ssRNA viruses, and pfam01728, a ribosomal RNA FtsJ-like MT, similar to the MTR domains found in flavivirus NS5 proteins. The polyprotein also contained HEL (pfam01443) and RdRp domains (pfam00978). The other two proteins encoded in the STNLV1 sequence did not show similarity with any viral proteins. In negeviruses, ORF2 and ORF3 are predicted to encode GPs and membrane proteins, respectively [44]. Two of the incomplete sequences belonged to STNLV2, which were most similar to the nege-like virus Wuhan house centipede virus 1 (YP_009342435.1) (Appendix A).

The four sequences belonging to soybean thrips jivi-like virus (STJLV) 1 and STNLV2 were most similar to members of the *Jivivirus* genus proposed to describe novel viruses from grapevine [45]. Members of the *Jivivirus* genus have tripartite genomes related to members of the *Virgaviridae*. The predicted aa sequence of STJLV1, RNA1 (MT240787) and STJLV1, RNA2 (MW039368) were most similar to the P1 and P2 proteins encoded by grapevine-associated jivivirus (GaJV) 2 with 44% and 43% aa sequence identity, respectively. The predicted aa sequences of STJLV2, RNA1 (MT240786) and STJLV2, RNA 2 (MW039369) were most similar to the P1 and P2 proteins encoded by GaJV1 with 55% and 65% aa sequence identity, respectively. Jivivirus P1 proteins contain MTR and HEL domains while P2 proteins contain an RdRp domain. The third segment, RNA3, encodes an N-terminal ATP-binding domain of a helicase similar to flavivirus NS3 [45]. We were unable to identify the third, flavi-like, RNA segments for these two viruses from our data. The assembled contigs also included four sequences from previously described plant viruses: RNAs 1, 2, and 3 of aphid-transmitted peanut stunt virus (PSV; MT293135, MT293136, and MT293137) and the genome of turnip vein-clearing virus (TVCV; MT293141) (Appendix A). 

Phylogenetic analysis based on RdRp sequences placed STVLV, STNLV, and STJLV sequences in different clades (Figure 4). These sequences clustered in two major groups with origins phylogenetically related to two plant virus families (*Kitaviridae* and *Virgaviridae*). STNLV1, STVLV1, and STVLV2 branched in a major group of arthropod-infecting/arthropod-borne viruses, which contained two subgroups. Subgroup1 included STNLV1 and other negeviruses and members of the *Kitaviridae* (genera; *Blunervirus*, *Cilevirus*, and *Higrevirus*), which supported the previous observations that negeviruses are closely related to plant infecting *Kitaviridae* members [43,44]. Furthermore, STNLV1 was associated with members of the proposed *Nelorpivirus* genus in the Negevirus taxon. STVLV1 and STVLV2 branched in Subgroup 2 with unclassified virga-like viruses reported from insects. STJLV1 and STVJV2 grouped with proposed jiviviruses in the second major clade that included members of the *Virgaviridae*, a family of plant-infecting viruses. The STNLV2 sequences were not included in the analysis because they lacked RdRp domains.

#### 3.2.4. Sequences Related to Members of the *Tolivirales*

Eleven novel sequences (named soybean thrips tombus-like virus (STTLV) 1 to STTLV11 resembling the genomes of unclassified tombus-like viruses from arthropods were assembled from the HTS data. These sequences were predicted to encode SF3 RdRps (pfam00998). The STTLV1 (MT240788) sequence contained two ORFs encoding an 83-kDa protein of unknown function from ORF1 and a 64-kDa RdRp from ORF2 (Figure 1). STTLV2 (MT240789), STTLV3 (MT240790), STTLV4 (MT240791), and STTLV5 (MT240793) contained four ORFs in which ORF1 and ORF4 encoded proteins (34–44 kDa and 17–23 kDa, respectively) of unknown functions. ORF2 was predicted to encode RdRp (60–71 kDa) and ORF3 was predicted to encode a CP of 24–28 kDa. The genome sequences of STTLV6 to STTLV11 (MT240794–MW023846, respectively) were 5’ and/or 3’-incomplete. The STTLV7 sequence encoded a structural protein that contained a conserved nodavirus capsid domain (pfam11729). The STTLV11 sequence contained a 64-kDa RdRp, which showed the highest aa sequence identities with plant viruses in the *Tombusviridae* family. We also discovered sequences (MT240792 and MW039367) from soybean thrips that were 95% identical to the genome sequences of Wuhan insect virus (WIV) 21 RNA1 (LC516847.1) and Hubei tombus-like virus (HTLV) 2 (NC_032965.1), respectively, which were originally reported from China [46]. 

In phylogenetic analysis with recognized members of the *Tombusviridae*, STTLV1 to STTLV11 formed two major groups (Figure 5). Except for sequences STTLV6, STTLV10, and STTLV11, all formed a major group with unclassified tombus-like viruses discovered from arthropods. STTLV6 and STTLV11 shared a monophyletic origin with plant viruses and branched into two separate clades with unclassified viruses. Bi-segmented unclassified insect tombus-like viruses such as WIV21 formed a separate branch.

#### 3.2.5. Sequences Related to Members of the *Amarillovirales*

Members of the *Flaviviridae* infect both vertebrates and invertebrates. The viruses that infect invertebrate hosts are exclusively members of the *Flavivirus* genus, which possess unsegmented genomes that encode single polyproteins that are cleaved into structural and nonstructural proteins [47]. A segmented flavi-like virus, Jingmen tick virus, was discovered from ticks [48]. Subsequently, additional segmented flavi-like viruses (Jingmen viruses) were reported from other arthropods and vertebrates [46], indicating their abilities to infect a broad range of hosts. Genomes of these viruses include four segments, two of which are predicted to encode proteins related to NS3 and NS5 proteins of *Flavivirus* members, suggesting that they are evolutionarily related [38]. The other two segments are predicted to encode proteins distinct to Jingmen viruses. 

Contigs representing six segmented flavi-like viruses were discovered from soybean thrips, five of which were novel virus-like sequences and named soybean thrips virus (STV) 1 to STV5. Four genome segments were discovered for STV2 (MW023854–MW023857) (Figure 1) and STV3 (MW033628–MW033631), three segments from STV1 (MW023851–MW023853) and STV4 (MW033624–MW033626) and one segment from STV5 (MW033632). Like other segmented flavi-like viruses, segment 1 of soybean thrips sequences contained an ORF encoding an NS5-like protein (102–108 kDa) that had two N-terminal MT domains (cd20761, pfam01728) and a flavivirus RdRp domain (pfam00972). Segment 2 encoded a protein (14–15 kDa) of unknown function and a putative GP (40–45 kDa). Segment 3 coded for a flavi-like NS3 protein (90–93 kDa), which contained both peptidase and helicase domains. Segment 4 was predicted to encode a CP (27–30 kDa) and a 50–65 kDa protein of unknown function. We also detected the four segments (MW023847–MW023850) of Wuhan aphid virus 1 (WAV1). The proteins encoded by the soybean thrips isolate of WAV1 shared 79% to 91% aa sequence identities with the corresponding proteins of the virus originally reported from China. We also detected a contig (MW039350) whose predicted aa sequence was 85% identical to an unsegmented flavi-like virus originally discovered from house fly, Shayang fly virus 4 (SFV4; YP_009179225.1) [38]. Although all genome segments could not be identified for the segmented flavi-like viruses, the discovery of these viruses provides valuable information on the molecular features of this group of viruses in thrips.

In phylogenetic analysis based on the predicted aa sequences of NS5-like proteins, soybean thrips flavi-like viruses grouped with members of the genus *Flavivirus* (Figure 6) in a large clade confirming their evolutionary relationship with this taxon. Although STV1 to STV4 branched in a separate clade, they shared a common monophyletic origin with Jingmen viruses. This also suggests that Jingmen viruses and soybean thrips flavi-like virus sequences are phylogenetically distinct, although they share similar segmented genome organizations and an evolutionary relationship to members of *Flaviviridae.*

#### 3.2.6. Sequences Related to Members of the *Tymovirales*

Three sequences related to members of the order *Tymovirales* were discovered in the assembled data (Appendix A). One contig represented a novel plant virus sequence, soybean carlavirus 1 (SCV1; MT293130), which we detected previously in RNA-Seq data of field-grown soybean leaves (submitted for publication). The RdRp and CP encoded by the soybean thrips SCV1 isolate were 96% and 100% identical with the aa sequences of the original isolate of SCV1 from soybean, but 62% and 84% identical, respectively, to the RdRp and CP aa sequences of red clover carlavirus A. A contig (MT240795) representing a North American isolate of Hubei macula-like virus 3 (HMLV3), which was originally reported from China [46], was also detected in the transcriptome data. The encoded proteins of this sequence showed 93%, 88.5%, and 56% aa identity with their orthologues in the HMLV3 isolate from China (KX883800.1). The predicted aa sequence of a third contig (MT293134) was 97% identical to red clover vein mosaic virus (RCVMV), which is transmitted by aphids. In phylogenetic analysis, SCV1 and RCVMV grouped with members of the family *Betaflexiviridae*, while the North American isolate of HMLV3 grouped with other invertebrate viruses related to members of the family *Tymoviridae* (Figure 7).

#### 3.2.7. Sequences Related to Members of the *Permutotetraviridae*

The predicted aa sequences of three contigs showed highest identity to virus-like sequences from other insects that resembled members of *Permutotetraviridae* (Appendix A). Viruses in the family *Permutotetraviridae* have monopartite genomes with two major ORFs, one encoding a permuted RdRp, and the other encoding a CP precursor [49]. Canonically, the catalytic motifs (A to D) are arranged in the order A–B–C–D in the palm subdomain of the active sites of RdRps of RNA viruses. In contrast, the order of the catalytic motifs is C–A–B–D in members of the *Permutotetraviridae*. A few members of the family *Birnaviridae* that have dsRNA genomes also possess permuted RdRps [49]. The soybean thrips permutotetra-like virus (STPTLV) 1 sequence (MT240780, 4290 nt) contained an RdRp (110 kDa) with a permuted motif arrangement (C–A–B–D) in which the GDD region is upstream of other palm subdomains. Unlike other *Permutotetraviridae* members, the STPTLV1 sequence encoded a third small 17-kDa protein of unknown function in addition to the 47-kDa CP (VP2; Figure 1). The other two permutotetraviridae-like sequences, STPTLV2 (MW039362) and STPTLV3 (MW039363), represented partial genome sequences. The VP2 of these viruses contained the shell (S) domain (pfam00729) of the CPs of *Birnaviridae* members. The predicted aa sequence of a fourth contig (MW033624) was 99.8% identical to Aphis glycines virus 2 (AGV2; YP_009179352.1), which also had a permuted RdRp [50]. 

In the phylogenetic analysis of predicted RdRp aa sequences, STPTLV1 grouped in a large diverse clade that included members of the family *Permutotetraviridae*, but was mostly populated by unassigned permutotetra-like virus sequences including Drosophila A virus (DAV; Figure 8A). To examine the taxonomic relationship of STPTLV1, STPTLV2, and STPTLV3, phylogenetic analysis was performed based on the VP2 sequences because the STPTLV2 and STPTLV3 sequences lacked RdRp domains. STPTLV1 and STPTLV2 were grouped with unassigned permutotetra-like virus sequences that again included DAV and showed stronger affinity with members of the bi-segmented *Birnaviridae* than the *Permutotetraviridae* (Figure 8B). The VP2 sequences of STPTLV3 and AGV2 shared a common phylogenetic origin with *Permutotetraviridae* members, but formed a separate clade with other unclassified permutotetra-like viruses from insects. The relationships among the CPs and the RdRps and the close phylogenetic relationship of viruses belonging to the *Birnaviridae* and *Permutotetraviridae* suggest evolutionary links between positive-sense ssRNA and dsRNA viruses of insects.

#### 3.2.8. Sequences Related to Members of the *Wolframvirales* and *Cryppavirales*

Predicted aa sequences of four contigs showed 31 to 44% identity to viruses in the phylum *Lenarviricota* (Appendix A). All four sequences encoded a single protein that contained a mitovirus RdRp domain (pfam05919) (Figure 1). With the possible exception of two putative bipartite narnaviruses, members of the families *Narnaviridae* (order *Wolframvirales*) and *Mitoviridae* (order *Cryppavirales*) have monopartite genomes of positive-sense ssRNA of about 3 kb, which contain a single ORF that encodes viral RdRp. These viruses have been reported from algae, apicomplexans, arthropods, fungi, insect, oomycetes, trypanosomatids, and yeast [51]. Narnaviruses and mitoviruses do not express structural proteins and thus exist as RNA-protein complexes either in the cytosol (members of *Narnavirus* genus) or in mitochondria (members of the *Mitovirus* genus) [52]. Since mitoviruses replicate in mitochondria, their genome sequences are translated using the mitochondrial genetic code in which UGA codons are translated as tryptophan rather than translation terminators. For two contigs (designated soybean thrips mito-like virus [STMLV] 1 [MT293144] and STMLV2 [MT293145]; Appendix A), the major ORF was translated only using the mitochondrial genetic code suggesting that these sequences represented mitovirus genomes. The remaining two sequences (soybean thrips narna-like virus [STNLV] 1 [MT293142] and STNLV2 [MT293143]; Appendix A) were translated with the universal genetic code. In phylogenetic analysis, STMLV1 and STMLV2 sequences grouped with recognized members of the *Mitoviridae*, and STNLV1 and STNLV2 sequences grouped with recognized members of the *Narnaviridae* (Figure 9). 

#### 3.2.9. Sequences Related to Members of the *Ourlivirales*

The predicted aa sequences of three contigs contained pfam05919 domains, which are present in the RdRps of members of the family *Botourmiaviridae*. Members of the family have monopartite or tripartite genomes in which each segment encodes a single protein [53]. The *Botourmiaviridae* currently includes four genera of which *Ourmiavirus* members are non-enveloped tripartite plant-infecting viruses. The genera *Botoulivirus*, *Magoulivirus*, and *Scleroulivirus* include non-encapsidated, monopartite, and fungal viruses encoding RdRps that show the closest similarity to those of ourmiaviruses. Some mycoviruses do not express CPs. Wang et al. [54] showed that a single RNA encoding an RdRp is sufficient for replication, infection, and transmission of ourmia-like viruses in fungi. The two ourmiavirus-like sequences identified in this study were isolates of a single virus, tentatively named soybean thrips ourmia-like virus (STOLV) 1 (MW039364 and MW039365) (Figure 1), which showed 89% identity in both nt and aa sequences of RdRp. Phylogenetic analysis including representative members of all genera in *Botourmiaviridae* showed that STOLV1 and STOLV2 (MW039366) shared a monophyletic origin with several unclassified ourmia-like mycoviruses that might represent a separate taxonomical group in the family (Figure 10). This group further branched in to two distinct clades supported with high bootstrap values with STOLV1 and STOLV2 in separate branches. 

### 3.3. Sequences Related to Negative-Stranded RNA Viruses

Fifty-four sequences discovered from soybean thrips metatranscriptome data represented both segmented and unsegmented negative-stranded RNA viruses. The most abundant of these sequences was SVNV, followed in decreasing order by soybean thrips thogotovirus 1 (STTV1), soybean thrips associated orthotospovirus 1 (STaTV1, soybean thrips associated tenui-like virus 1 (STaTLV1), and soybean thrips rhabdo-like virus (STRLV) 1 (Appendix A)

#### 3.3.1. Sequences Related to Members of the *Articulavirales*

Fifteen contigs resembled the genome segments of negative-stranded RNA viruses in the genera *Thogotovirus* and *Quaranjavirus* in the family *Orthomyxoviridae*, and putatively represented six novel viruses (Appendix A). These sequences were tentatively named STTV1 and STTV2. The remaining four sequences (soybean thrips quaranja-like virus [STQLV] 1 to STQLV4) were phylogenetically related to the members of the genus *Quaranjavirus* (Appendix A). Thogotoviruses and quaranjaviruses are enveloped tick-borne viruses that infect humans, domestic animals, birds, and arthropods [55,56]. Genomes of recognized members of these two genera contain six RNA segments that encode, respectively, the polymerase complex (PB2, PB1, and PA), GP, nucleoprotein (NP), and matrix protein (MP).

The genome of STTV1 showed 33–54% aa identity with the corresponding segments of Hubei orthoptera virus 6 (KX883884–KX883888), an unclassified negative-stranded RNA virus (Appendix A). Five RNA segments were discovered for STTV1 (MT195539–MT195543). Each segment contained a single ORF (Figure 1). STTV1 segment 1 encoded an 86-kDa PB2 polymerase, segment 2 an 80-kDa PB1 polymerase, segment 3 a 75-kDa PA polymerase, segment 4 a 57-kDa GP, and segment 5 a 52-kDa NP. We obtained only segment 4 (MW033633) of STTV2 from the assembled data that encoded a 54 kDa GP. The partial genomes obtained for quaranjaviruses included segments 1 (MW033634), 2 (MW033635), 3 (MW033636), and 5 (MW033637) for STQLV1, segments 2 (MW033638), 3 (MW033639), and 4 (MW033640) for STQLV2, and segments 2 (MW033641) and 4 (MW033642) for STQLV3. In cases of sequences where the complete aa sequences were obtained for the encoded proteins, the size ranged from 91 kDa for PB2 polymerase, 91–93 kDa for PB1 polymerase, 82–88 kDa for PA polymerase, and 62 kDa for NP. In the phylogenetic analysis based on PB1 polymerase (RdRp), STTV1 formed a separate clade with Hubei orthoptera virus 6 among other recognized thogotoviruses and might represent a new taxonomical group in the genus. STQLV1, STQLV2, and STQLV3 were grouped with other recognized members of the *Quaranjavirus* genus (Figure 11).

#### 3.3.2. Sequences Related to Members of the *Bunyavirales*

Twenty contigs in the assembled data were related to members of the families *Arenaviridae*, *Bunyaviridae*, *Nairoviridae*, *Phasmaviridae*, *Phenuiviridae*, or *Tospoviridae* in the order *Bunyavirales*, 16 of which had not been reported previously (Appendix A). Members of the *Bunyavirales* have tripartite genomes consisting of large (L), medium (M), and small (S) RNA segments with conserved terminal sequences among viruses in each genus [57]. The L segment encodes an RdRp and the M segment encodes the viral GPs, Gn, and Gc. In most *Bunyavirales* members, the L and M segments are negative stranded with some exceptions where the M segments are ambisense (e.g., *Orthotospovirus*). In the genus *Orthotospovirus* members (family, *Tospoviridae*), the Gn and Gc are encoded from the negative strand, while a nonstructural protein, NSm, which is the putative movement protein in plant-infecting viruses, is encoded from the positive strand. The S segment is ambisense in most *Bunyavirales* members and encodes the NP from the negative strand and a nonstructural protein (NSs) from the positive strand. 

Among the bunyavirus-like sequences from soybean thrips, five were predicted to be RdRp-encoding L segments: soybean thrips bunya-like virus (STBLV) 1 (MT224143; 342 kDa RdRp), STBLV2 (MW033651; partial RdRp), STBLV3 (MT224145; 240 kDa RdRp), STBLV6 (MW033651; partial RdRp), and STBLV7 (MW033652; partial RdRp) (Appendix A). Sequences STBLV4 (MT293149) and STBLV5 (MW023862) represented M segments encoding GP precursors (152 kDa and 168 kDa, respectively). Among the remaining bunyavirus-like sequences, three were putative members of the *Phenuiviridae* that included two segments (RNA1 and RNA4) of a possible new member of the genus *Tenuivirus*, which we tentatively designated STaTLV1. The RNA1 segment of STaTLV1 (MT224144) encoded a 318 kDa RdRp and the RNA4 (MW033650) encoded a 20 kDa major non-capsid protein and a 33 kDa nonstructural protein (NS4) in opposite polarity (Figure 1). The predicted aa sequences of these contigs showed 27–30% aa sequence identity to proteins encoded by plant tenuiviruses (Appendix A). STBLV8 (MW033653) encoded a partial GP precursor that contained conserved domain of phlebovirus GP G2 (pfam07245). Two other sequences were related to members of the *Nairoviridae* family and encoded proteins that contained a nairovirus nucleocapsid domain (pfam02477). These two sequences shared 82% and 88% nt and aa sequence identity, suggesting that they are isolates of a single virus that were provisionally named STBLV9. Two contigs, designated soybean thrips negative-stranded RNA virus [STNRV] 2), represented L (MT224152) and S (MW033657) segments (encoding 393-kDa L and 46-kDa NP, respectively) of a novel bunya-like virus that showed significant homology to a member of the family *Arenaviridae*. Consistent with that association, the STNRV2 L segment contained an arenavirus RdRp domain (pfam06317) (Figure 1). The remaining six of the bunyavirus-like sequences belonged to the members of *Tospoviridae* that includes all three segments of SVNV (MT293138–MT293140), partial S segment of TSWV (MW033649), and L (MT195544) and M (MT195545) segments of a novel orthotospovirus, tentatively named STaTV1 (Appendix A). The L segment of STaTV1 encoded a 334-kDa L protein and the M segment encoded two proteins in opposite polarity, a 130-kDa GP and a partial sequence of a nonstructural protein (Figure 1). 

In phylogenetic analysis, the soybean thrips bunyavirus-like sequences grouped in the *Bunyavirales.* STaTV1, STaTLV1, and STNRV1 grouped with recognized members of the *Tospoviridae*, *Phenuiviridae*, and *Arenaviridae*, respectively (Figure 11). STBLV1 grouped with unclassified bunya-like viruses, and STBLV3 branched with *Phasmaviridae* members. The partial sequences of STBLV2, STBLV6, and STBLV7 were not included in the analysis because they lacked conserved RdRp domains.

#### 3.3.3. Sequences Related to Members of the *Jingchuvirales*

Nine sequences (Appendix A) were related to members of the family *Chuviridae* in the order *Jingchuvirales*, two of which were segments of a virus that was tentatively named soybean thrips chu-like virus 1 (STCLV1). Most members of the *Chuviridae* are arthropod-associated viruses in the genus *Mivirus* and have negative-stranded RNA genomes [58]. They occupy a phylogenetically intermediate position between segmented and unsegmented negative-stranded RNA viruses and have diverse genome organizations, from linear to circular forms that can be unsegmented or bi-segmented [59]. *Chuviridae* members also exhibit a diverse order of genes in their genomes. The general arrangement of genes in linear unsegmented chuvirus genomes is GP-NP-L. In most of the circular chuviruses, genes are arranged in the order L-GP-NP [58]. Additionally, some chuvirus genomes lack the GP coding region. In bi-segmented chuviruses, the L segment encodes RdRp and the S segment encodes the putative GP, NP, and possibly a protein of unknown function. 

All chuvirus-like genome sequences obtained from soybean thrips were predicted to be linear. The L segment (MT224150) of the bi-segmented STCLV1 encoded a 255 kDa RdRp and partial sequence of S segment (MT293148) encoding a 35 kDa hypothetical protein and a partial GP (Figure 1). The S segment (MT293147) for STCLV2 contained a 75-kDa GP, 48-kDa NP, and a 14-kDa hypothetical protein. The remaining STCLV contigs represented partial genome sequences in which the complete coding sequences of a protein was obtained only for GP in STCLV3 (MT293151) and STCLV4 (MW033643). In phylogenetic analysis based on RdRp sequences, STCLV1 grouped with members of the *Jingchuvirales* (Figure 11). An additional analysis based on GP sequences was conducted since the STCLV2 to STCLV8 sequences lacked RdRp. All of the STCLV sequences branched together in the tree with other chuviruses, confirming their taxonomical status (Figure 12).

#### 3.3.4. Sequences Related to Members of the *Serpentovirales*

Sequences STNRV1 (MT224151), STNRV3 (MT224153), STNRV6 (MT293150), STNRV7 (MW039378), and STNRV8 (MW039379) were phylogenetically related to members of the *Ophiovirus* genus, family *Aspiviridae*, in the order *Serpentovirales* (Appendix A). Members of *Aspiviridae* are filamentous plant viruses and have segmented genomes with three or four segments. The family *Aspiviridae* contains seven recognized members, four of which are transmitted by soil-borne fungi [60]. The largest segment of the genomes of these viruses encodes an RdRp and a small protein. The sequences from soybean thrips encoded only RdRps (Figure 1) and did not group with the recognized members of the *Ophiovirus* genus in the phylogenetic analysis although they shared a common monophyletic origin. Instead, they branched with unclassified, negative-stranded RNA mycoviruses [61,62] among which soybean thrips viruses formed a separate clade (Figure 11). The phylogenetic grouping suggest that these sequences might represent genome segments of mycoviruses.

#### 3.3.5. Sequences Related to Members of the *Mononegavirales*

Five of the novel virus-like sequences resembled the genomes of viruses that have monopartite genomes in the order *Mononegavirales* (Appendix A). Three sequences, STRLV1 (MT224147), STRLV2 (MT224148), and STRLV3 (MW023861), showed highest identities to the members of *Rhabdoviridae*, one of the 11 families in *Mononegavirales* [63]. Rhabdoviruses infect vertebrates, plants, and arthropods and generally encode five proteins in their monopartite/bipartite genomes [63]. Plant-infecting members of the *Rhabdoviridae* are transmitted by arthropod vectors such as aphids, planthoppers, leafhoppers (*Cytorhabdovirus* and *Nucleorhabdovirus*), *Brevipalpus* mites (*Dichorhavirus*), and the soil-borne fungus, *Olpidium brassicae* (*Varicosavirus*) [64]. No members of the family have been reported to be transmitted by thrips or to infect thrips. 

In monopartite members, rhabdovirus proteins are encoded in the order NP, phosphoprotein (P), MP, GP, and L, while in bipartite members (genera, *Varicosavirus* and *Dichorhavirus*) the L protein is encoded from a separate genome segment. In some *Rhabdoviridae* members, additional small ORFs are present [63]. STRLV1 and STRLV2 sequences encoded NPs of 49 and 48 kDa and L proteins of 244 and 242 kDa, respectively (Figure 1). The three other proteins encoded between NP and L proteins (25 kDa, 20 kDa, 51 kDa for STRLV1 and 22 kDa, 25 kDa, and 53 kDa for STRLV2) did not show significant similarity with other rhabdovirus proteins. The genome of STRLV3 (Figure 1) encoded 60-kDa NP, 245-kDa L, and two hypothetical proteins (95 kDa and 30 kDa) between NP and L. The partial sequence obtained for the fourth sequence, STNRV4 (MT224149) encoded three proteins, a partial hypothetical protein, 79-kDa GP, and a partial RdRp. The fifth novel virus sequence, STNRV5 (MW033648), contained only a partial RdRp. 

In phylogenetic analysis, STRLV1, STRLV2, and STRLV3 grouped with other arthropod-borne members of the family *Rhabdoviridae* (Figure 11), indicating the possibility that they are either thrips-infecting viruses or viruses from other plant feeding arthropods circulating in soybean thrips. Although STNRV4 shared a monophyletic origin with members of *Nyamiviridae*, it formed a separate clade with Hubei diptera virus 11 (NC_033055.1), an unclassified negative-stranded RNA virus and may represent a new taxon in the *Mononegavirales*. STNRV5 was not included in the analysis as it did not encode a conserved RdRp domain.

### 3.4. Sequences Related to dsRNA Viruses 

Twenty-two of the sequences from soybean thrips were related to dsRNA viruses in the families *Partitiviridae* (order *Durnavirales*), *Reoviridae* (order *Reovirales*), and *Totiviridae* (order *Ghabrivirales*). 

#### 3.4.1. Sequences Related to Members of the *Durnavirales*

The recognized members of the *Partitiviridae* have monocistronic, bi-segmented dsRNA genomes that encode only RdRp and CP [65]. However, some unclassified partiti-like viruses have multisegmented genomes (e.g., Aspergillus flavus partitivirus 1 (three segments), Atrato partiti-like virus 1 (four segments), and Botryosphaeria dothidea virus 1 (five segments)). Fourteen partitivirus-like sequences were identified in the assembled data. One contig (MW039370) represented RNA2 of Wuhan fly virus 5 (YP_009342459.1) with 94% aa sequence identity (Appendix A). Eleven of the sequences were RNA1 segments that contained partial RdRps (domain cd01699), which were named soybean thrips partiti-like virus (STPaLV) 1 to STPaLV11 (MT648422–MT648430, MW039371, and MW023858). The remaining two sequences (MW023859 and MW023860) represented segments 2 and 3 of STPaLV11, which encoded a 54-kDa CP and a 54-kDa protein of unknown function, respectively (Figure 1).

Members of each genus in *Partitiviridae* have a unique host range. Members of *Alphapartitivirus* and *Betapartitivirus* infect fungi or plants. Members of the *Gammapartitivirus*, *Deltapartitivirus*, and *Cryspovirus* genera infect fungi, plants, and protozoans, respectively [65]. In the phylogenetic analysis of predicted RdRp aa sequences, STPaLV1, 3, and 8 grouped with *Betapartitivirus* members (Figure 13). The remaining sequences grouped with unclassified members of the family. These suggested that the partitivirus-like sequences discovered from soybean thrips are phylogenetically divergent and may represent viruses that infect fungi, parasites, or plants related to the soybean thrips life cycle. 

#### 3.4.2. Sequences Related to Members of the *Ghabrivirales*

Genomes of most members of the family *Totiviridae* contain two large overlapping ORFs in which the 5’-proximal ORF encodes the major CP and the 3’-proximal ORF encodes an RdRp. In viruses with overlapping ORFs, the RdRp is expressed only as a CP-RdRp fusion protein by a −1 frameshift [66]. We discovered five totivirus-like sequences from the assembled data. The sequence designated soybean thrips-associated totivirus (STaToV) 1 (MT293124) contained two ORFs that overlapped by just seven nucleotides (Figure 1), while the two ORFs of STaToV2 (MT293125) overlapped by 316 nt. STaToV1 encoded a CP of 84 kDa, which is typical for fungus-infecting totiviruses while STaToV2 encoded a much larger CP of 151 kDa, which is similar in size to the CP of arthropod-infecting totiviruses [67,68]. In both sequences, ORF2 contained a pfam02123 RdRp domain. Two sequences, soybean thrips-associated dsRNA virus (STaDRV) 1 (MT293146) and STaDRV2 (MW039372), represented partial genomes containing RdRps. The sequence STaDRV4 (MW033658) was predicted to encode a proline-alanine rich protein. These sequences grouped with different recognized taxa of the *Totiviridae* (Figure 14). STaToV1 was evolutionarily most closely related to members of the *Victorivirus* genus whose members infect primarily fungi. STaToV2, STaDRV1, and STaDRV2 grouped in two different clades with unclassified toti-like viruses thought to have arthropod hosts.

#### 3.4.3. Sequences Related to Members of the *Reovirales*

Three contigs (MT648431, MW039372, and MW039373) representing partial segments of a reo-like virus were tentatively named STaDRV3 (Appendix A). In phylogenetic analysis with the RdRp sequences of recognized and putative members of *Reoviridae*, STaDRV3 grouped with members of the subfamily *Spinareovirinae*, but in a separate clade with Shelly beach virus, another unclassified reo-like virus discovered from ticks in Australia [69] (Figure 15). 

### 3.5. Sequences Related to Single-Stranded DNA Viruses 

The transcriptome data contained two densovirus-like sequences, which were provisionally named soybean thrips denso-like virus (STDLV) 1 (MT240783) and STDLV2 (MW039377). Densoviruses broadly encompasses viruses in the family *Parvoviridae* and order *Piccovirales* that infect both vertebrate and invertebrate hosts and have linear, ssDNA genomes of 4 to 6 kb [70]. The family *Parvoviridae* is divided into three subfamilies: *Densovirinae*, *Hamaparvovirinae*, and *Parvovirinae*. The STDLV1 sequence contained three overlapping ORFs in which the first two encoded nonstructural proteins (NS1 [87 kDa] and NS2 [39 kDa]), and the third ORF encoded a 43-kDa structural protein (Figure 1). STDLV2 is an ambisense sequence that contained four ORFs (ORFs1, 2, 4, and 5) on the plus strand and a fifth, ORF3, on the complementary strand. Among these, ORFs 1 and 5 encoded a protein similar to NS1 proteins, and ORF5 contained a conserved domain of parvo NS1 SF (pfam01057). As densoviruses employ a rolling-circle replication strategy, it is possible that ORFs 4 and 5 are circularly permuted portions of ORFs 1 and 2 generated during library construction. In the phylogenetic analysis based on the NS1 (replicase) region, STDLV1 and STDLV2 grouped with members of the subfamily *Densovirinae* (Figure 16).

## 4. Discussion

Soybean thrips are common insect pests of soybean in the U.S. Even though feeding injury from soybean thrips is usually not economically damaging, it is one of the most significant vectors for SVNV. Virus diversity in other insect vectors/agricultural pests have been reported [14,16,18,19,21,71,72]. Our study, for the first time, revealed the remarkable virus diversity that exists in soybean thrips.

Analysis of metatranscriptome of soybean thrips collected from the central U.S. detected large numbers of diverse types of described and novel arthropod, fungal, and plant-infecting viruses. Of the 181 virus-like sequences discovered, 155 (86%) were novel sequences of which 123 are predicted to belong to putative members of arthropod-infecting virus taxa (Appendix A). Among the rest, six sequences are related to plant infecting members, 22 are related to mycoviruses, and four sequences are predicted to be genome segments of an orthotospovirus and a tenuivirus that infect both thrips and plants. Since there was no previous information on thrips virome, the novel sequences discovered in our study represent the first of those taxa in thrips. The 26 previously described virus-like sequences included 16 arthropod-infecting, and six plant-infecting sequences, and four orthotospovirus genome segments, which replicate in arthropods and plants.

Additionally, this study obtained greater than 90% coverage of genomes or genome segments of 104 new and 20 previously described virus genomes (Appendix A). The sequences discovered were highly diverse in their types of genetic material of which the majority were positive-sense ssRNA viruses (56.9%), followed in decreasing order by negative-stranded ssRNA viruses (29.8%), dsRNA viruses (13.3%), and ssDNA viruses (1.1%). These sequences represented viruses with monopartite or multipartite genomes. 

The positive-sense RNA viruses discovered from soybean thrips were phylogenetically related to arthropod, fungal, and plant-infecting members of the orders *Amarillovirales*, *Cryppavirales*, *Martellivirales*, *Ourlivirales*, *Picornavirales*, *Sobelivirales*, *Tolivirales*, *Tymovirales*, and *Wolframvirales*. The most abundant group of viruses were putative members of the *Iflaviridae* with just over 50% of the virus-derived Illumina sequence reads aligning to contigs showing phylogenetic affinity to members of the family followed by sobemovirus-like sequences (26%) (Appendix A). The negative-stranded RNA viruses discovered were related to the members of *Articulavirales*, *Bunyavirales*, *Jingchuvirales*, *Mononegavirales*, and *Serpentovirales*. The dsRNA viruses represented putative members of the *Durnavirales*, *Ghabrivirales*, and *Reovirales*. Although this was a transcriptome analysis, we discovered two densovirus-like sequences, putative members of the order *Piccovirales*. The phylogenetic topology (Figure 2, Figure 3, Figure 4, Figure 5, Figure 6, Figure 7, Figure 8, Figure 9, Figure 10, Figure 11, Figure 12, Figure 13, Figure 14, Figure 15 and Figure 16) and the availability of complete coding regions of multiple genomes indicate that many of these represent potentially new taxonomical groups in the orders above-mentioned. Phylogenetic analyses of these novel viruses give insights into the diverse nature and evolution of these sequences. 

Some of the novel arthropod-infecting viruses discovered from thrips have unique genome features. For example, although STPiLV1 grouped with dicistroviruses in phylogenetic analysis, it encoded a single polyprotein instead of two with a dicistro-like arrangement of structural and nonstructural protein domains. In contrast, STBV1 had a bicistronic monopartite genome with a picorna-like arrangement of structural and nonstructural polyproteins. In addition, six novel putative *Flaviviridae* members discovered from soybean thrips had segmented genomes similar to Jingmen viruses, a group of segmented flavi-like viruses.

Interestingly, some of the novel arthropod viruses showed highest aa identity to arboviruses that are transmitted by ticks. Examples are soybean thrips thogotoviruses, quaranjaviruses, and bunya-like viruses (STBLV1 and STBLV4) (Appendix A). Two negevirus-like sequences (STNLV1 and STNLV2) were also discovered in the soybean thrips data. Negeviruses are primarily reported from mosquitoes and have worldwide distribution and broad host ranges among dipterans [43]. Considering their broad host range in other dipterans, it is possible that these viruses are indeed thrips-infecting viruses. Another possibility is that these are mosquito viruses acquired by thrips while feeding on contaminated plant surfaces. 

Our results also revealed the existence of arthropod viruses with broad geographical ranges. AGV2, ALPV, RhPV, HAV1, HMLV3, HPiLV55, HTLV2, Pernambuco virus, SFV4, WAV1, WFV5, and WIV21 are previously reported arthropod-related viruses that have been detected in soybean thrips in which all except for the flavi-like viruses (WAV1 and SFV4) were at least 90% identical to the previously described sequences. HAV1, HMLV3, HPiV55, HTLV 2, WAV1, WIV21, SFV4, and WFV5 were reported originally from Hubei Province in China [38,46] and Pernambuco virus was reported from Brazil [34]. The high levels of nucleotide sequence identity among virus isolates from North America, Brazil, and China may be the result of the frequent exchange of insects between the continents [73,74]. The abundance of sequence reads for some of these viruses suggests that they replicate within soybean thrips and may indicate that these viruses have broad host ranges, which may have facilitated their spread across broad geographic ranges. 

Sequences discovered from thrips also included viruses that replicate only in plants that are transmitted by biological vectors or other means including LTSV, PSV, RCVMV, and TVCV. Additionally, four new sequences grouped with the plant viruses in the phylogenetic analysis (SCV1, STJLV1, STJLV2, and STSLV11) and may represent plant infecting viruses. LTSV was reported in North America only from Canada [40,41]. The discovery of these plant viruses in soybean thrips along with other new putative plant viruses highlights the importance of molecular surveillance programs in biological vectors to identify emerging plant viruses. 

In addition, we detected viruses that infect and replicate in both plants and their arthropod vectors (orthotospoviruses and tenuiviruses): SVNV, STaTV1, STaTLV1, and TSWV. Sequence reads for these viruses were highly abundant in the soybean thrips transcriptome (Appendix A) supporting the conclusion that these viruses infect and propagate in thrips and are not just surface or gut contaminants picked up during metagenomics studies. Thrips are the only insects that vector tospoviruses [3]. The novel orthotospovirus-like sequences, STaTV1, detected in this study have not been reported from soybean. Additional research is needed to determine the host range of the virus. The recognized members of tenuiviruses infect only monocots and are transmitted by plant hoppers [75]. Soybean thrips have been reported from multiple species of plants in the U.S. [9,10,11,12,13]. Therefore, the presence of viruses that infect other plant species or viruses that are not known to be transmitted by thrips was expected. In addition, plants have been shown to act as reservoirs of invertebrate picorna-like viruses [76]. 

Virus sequences obtained from metatranscriptomics not only represent the virome of the vector organism, but also include viruses of associated organisms. Some of the sequences discovered in this study could be viruses of symbionts and parasites associated with thrips. Alignment of the assembled contigs to reference protein sequences showed that 0.5% of the contigs aligned to parasitoid wasps, which could indicate the presence of parasitoids in the sampled thrips. The transcriptome data contained six novel putative mycovirus-like sequences belonging to three major lineages of positive-sense ssRNA viruses of eukaryotes in the families *Botourmiaviridae*, *Narnaviridae*, and *Mitoviridae*. We also found monocistronic negative-stranded ssRNA mycovirus-like sequences (order *Serpentovirales*, Appendix A) that encode only RdRps. There were also several novel dsRNA virus-like sequences belonging to the orders, *Durnavirales* and *Ghabrivirales*, which might represent mycovirus genomes. All these viruses shared a common phylogenetic origin with other mycoviruses of their respective taxa (Figure 9, Figure 10, Figure 11, Figure 13 and Figure 14), suggesting that these viruses are potentially viruses of fungal parasites of soybean thrips, soybean, or other fungal contaminants. 

The viruses discovered exhibited a huge variation in the abundance of sequence reads. This variation could be due to the difference in stage of infection at the time of thrips collection or incidence of infection in the collected thrips. It is also possible that some viruses of low abundance do not infect and replicate within thrips, but infect other arthropods and are acquired when thrips feed on virus contaminated plants or are viruses of plants that circulate through thrips while feeding on plants. Finally, it is possible that some partial and low abundance contigs could be derived from virus-like sequences integrated into the soybean thrips genome. Virus-like sequences have been detected in the genomes of multiple arthropod species [77,78,79,80,81]. Most frequently, the detected sequences are related to parvoviruses and GPs of negative-stranded RNA viruses [82]. Detection of sequences capable of encoding proteins related to those of positive sense ssRNA viruses are much less common, although flavivirus-like sequences have been detected in *Aedes* spp mosquito genomes [83]. BLAST searches of the genomes of three species related to soybean thrips, *Aptinothrips rufus* (GCA_902196195.1), *F. occidentalis* (GCA_000697945.4), and *Thrips palmi* (GCA_012932325.1) identified sequences related to GPs of negative-stranded RNA viruses, but no virus-like RdRP sequences were detected (data not shown). Even though endogenous virus-like sequences are often truncated and contain termination codons [81], most of the sequences assembled in this study represented nearly complete virus genomes of positive-sense ssRNA viruses with long ORFs, which suggests that most represent sequences of freely replicating viruses.

Viruses that infect and multiply in arthropod vectors can affect the feeding behavior, fecundity, and lifespan of their hosts [84,85,86]. In SVNV-infected soybean thrips, accumulation of high levels of virus was found to lower viability of thrips although infected females produced significantly more offspring compared with non-infected females [8]. Additionally, SVNV-infected thrips preferred to feed on non-infected leaflets compared with infected leaflets. Studies with TSWV suggested that the virus can influence the developmental time, mating behavior, fecundity, and sex determination of offspring in its vector *F. occidentalis* to facilitate its transmission [87]. Similarly in aphids, viruses have been shown to affect developmental time and reproduction [88]. Implications of the presence of the diverse viruses in soybean thrips, arthropod specific as well as fungal and plant viruses in its lifecycle, feeding behavior, and the vector potential are not known at this point.

## 5. Conclusions

The use of virus metagenomics has rapidly expanded our understanding of the diversity, ecology, evolution, and taxonomy of viruses. This study extended our knowledge of the unexplored and rapidly expanding world of insect viromes that provides valuable information on the prevalence, diversity, and evolution of viruses associated with thrips. The discovery and characterization of insect viruses is important in increasing our knowledge and understanding of insect–virus diversity, insect–virus interactions and their coevolution, and insect immune defenses. Along with previous studies [38,46,58,89], our findings reaffirm the vast diversity of viruses in arthropods and suggests that insects could be reservoirs of known and novel viruses including plant-infecting viruses. In addition, the discovery of novel plant-infecting viruses highlights the potential use of VEM along with vector surveys to identify emerging viruses of agriculturally important crops present in a geographic region.

## Figures and Tables

**Figure 1 viruses-12-01376-f001:**
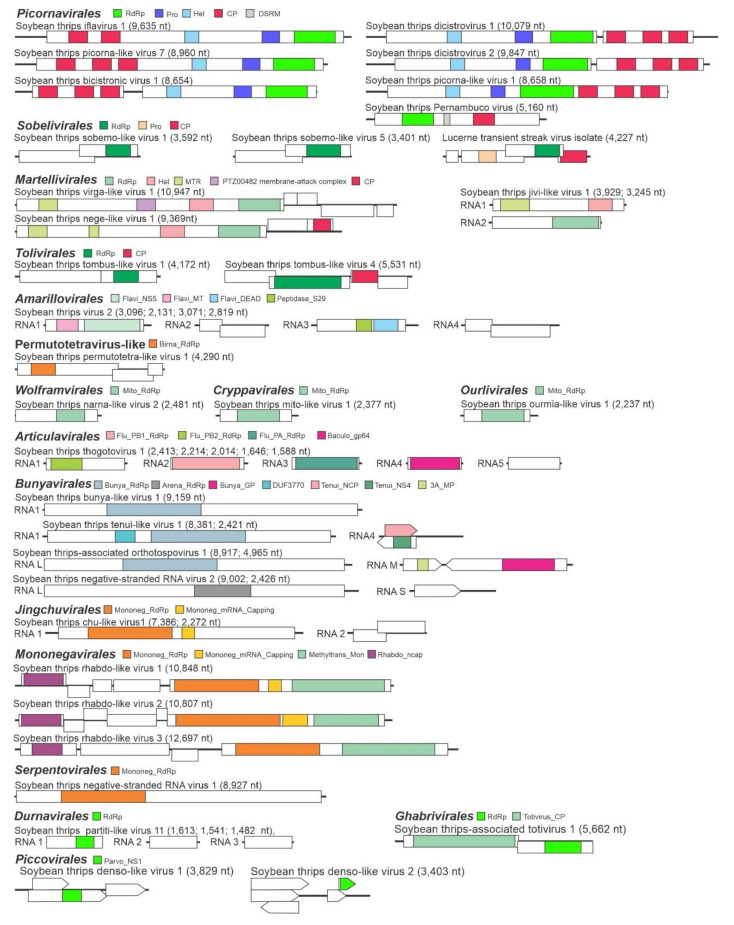
Examples of genome organizations of virus-like sequences recovered from soybean thrips transcriptome assembly. Open reading frames are represented by boxes that are staggered to indicate the different reading frames occupied. Colored boxes indicate conserved domains: RdRp = RNA-dependent RNA polymerase; Pro = protease; DSRM = double-stranded RNA binding motif (PF00035); Hel = helicase; MTR = methyltransferase; CP = capsid protein; Flavi_NS3 = *Flavivirus* RdRp subunit NS3 (PF00972); Flavi_NS5 = *Flavivius* NS5 motif; Flavi_MT FtsJ-like methyltransferase (PF01728); Flavi_DEAD = *Flavivirus* DEAD domain helicase (PF07652); Peptidase_S29 = Hepatitis C virus NS3 protease (PF02907); Mito_RdRp = Mitovirus RdRp (PF05919); Mononeg_RdRp = *Mononegavirales* RdRp (PF00946); Mononeg_mRNA= *Mononegavirales* mRNA-capping region V; Methyltrans_Mon = Virus-capping methyltransferase (PF14314); Parvo_NS1 = *Parvovirus* nonstructural protein NS1 (PF01057); Flu_PB1_RdRp = Influenza RdRp subunit PB1 (PF00602); Flu_PB2_RdRp = Influenza RdRp subunit PB2 (PF00604); Flu_PA_RdRp = Influenza RdRp subunit PA (PF00603); Baculo_gp64 = Baculovirus gp64 envelope glycoprotein (PF03273); Arena_RdRp = *Arenavirus* RdRp (PF06317); Totivirus_CP = *Totivirus* CP (PF05518); Rhabdo_ncap = *Rhabdovirus* nucleocapsid protein (PF00945); Birna_RdRp = *Birnavirus* RdRp (PF04197); Bunya_RdRp = *Bunyavirus* RNA dependent RNA polymerase (PF04196); Bunya_GP = *Bunyavirus* glycoprotein G1 (PF03557); DUF3770 Protein of unknown function (DUF3770); Tenui_NCP = Tenuivirus major non-capsid protein (PF04876); Tenui_NS4 = *Tenuivirus* movement protein, NS4 (PF03300); 3A_MP = 3A/RNA2 movement protein family (PF00803).

**Figure 2 viruses-12-01376-f002:**
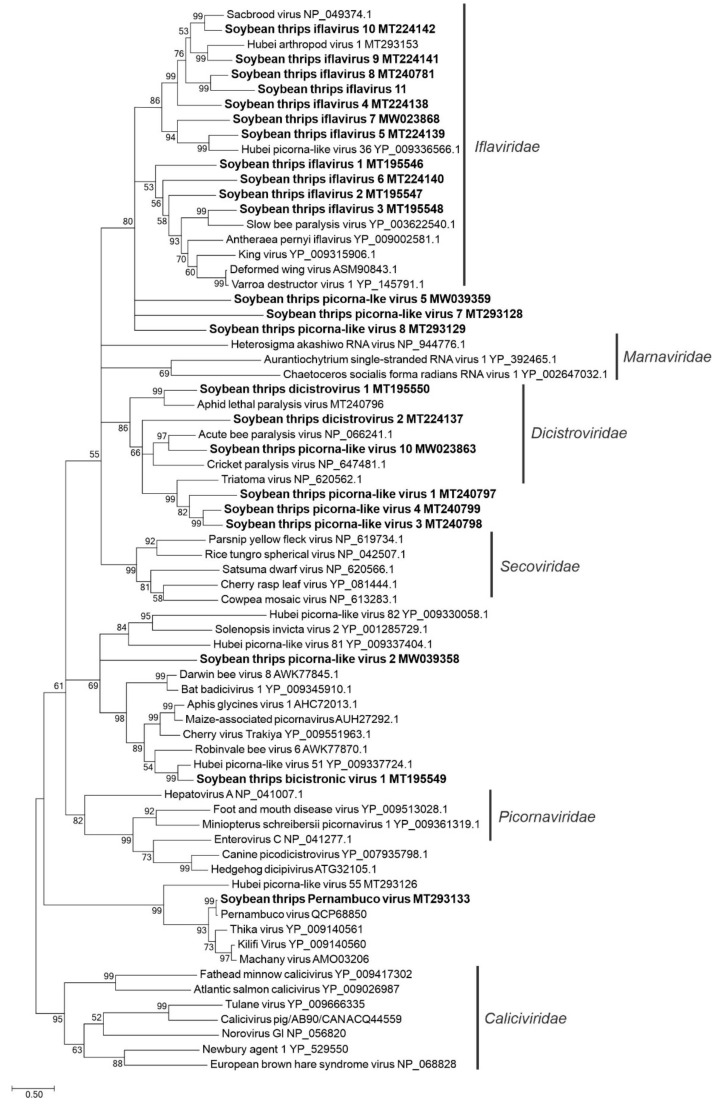
Phylogenetic analysis of the predicted RNA-dependent RNA polymerase amino acid sequences encoded by contigs assembled from soybean thrips RNA-seq data (indicated in bold) and related viruses in the order *Picornavirales*. Predicted amino acid sequences containing RNA-dependent RNA polymerase domains were aligned using MUSCLE. Phylogenetic analyses were performed using the maximum likelihood method in MEGA7. Numbers at nodes indicate percent bootstrap support (500 replicates). Nodes with less than 50% bootstrap support were collapsed to the next higher level. GenBank accession numbers are indicated after each sequence name.

**Figure 3 viruses-12-01376-f003:**
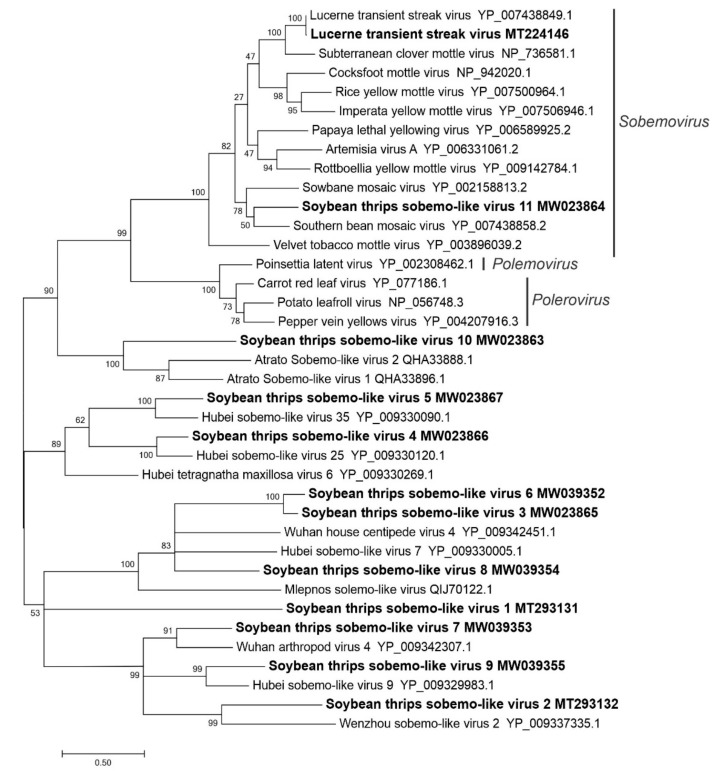
Phylogenetic analysis of the predicted RNA-dependent RNA polymerase amino acid sequences encoded by contigs assembled from soybean thrips RNA-seq data (indicated in bold) and related viruses in the order *Sobelivirales*. See Figure 2 legend for details.

**Figure 4 viruses-12-01376-f004:**
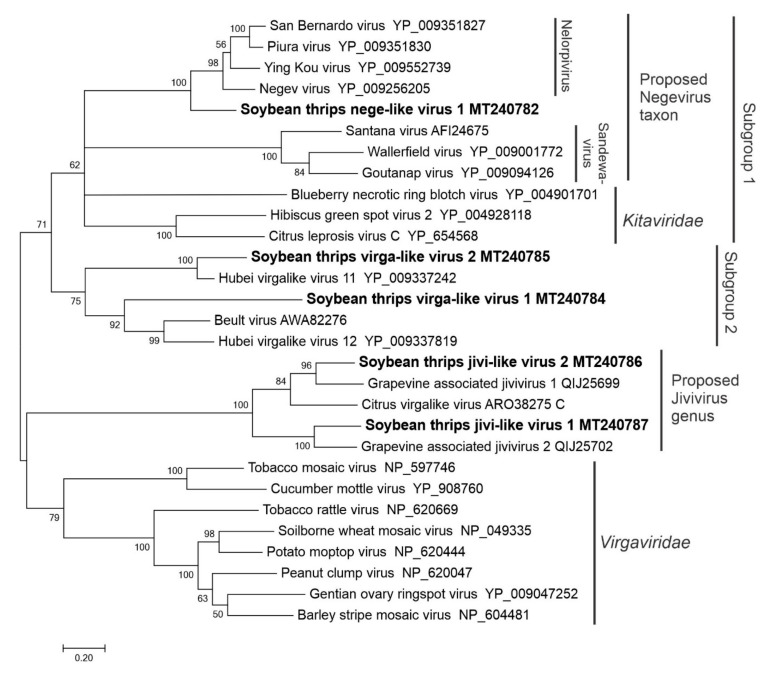
Phylogenetic analysis of the predicted RNA-dependent RNA polymerase amino acid sequences encoded by contigs assembled from soybean thrips RNA-seq data (indicated in bold) and related viruses in the order *Martellivirales*. See Figure 2 legend for details.

**Figure 5 viruses-12-01376-f005:**
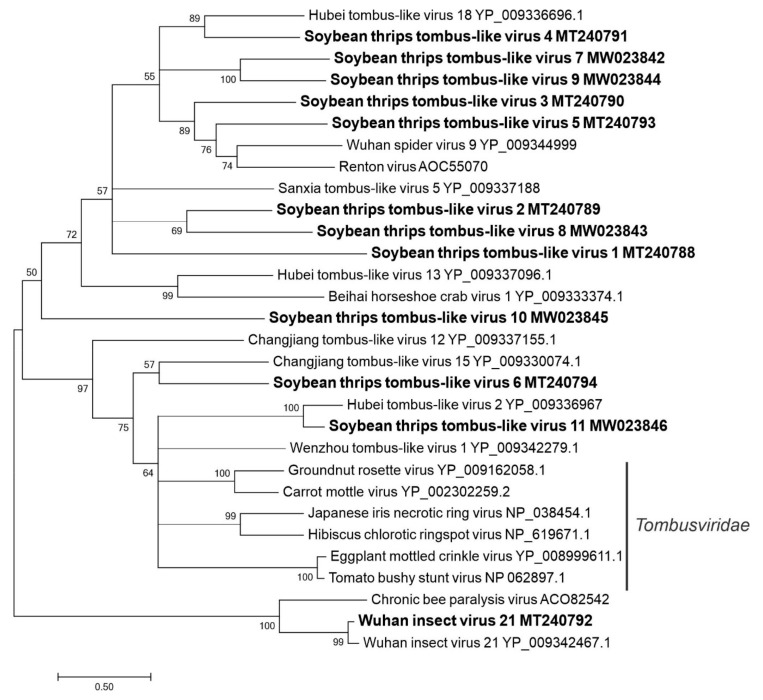
Phylogenetic analysis of the predicted RNA-dependent RNA polymerase amino acid sequences encoded by contigs assembled from soybean thrips RNA-seq data (indicated in bold) and related viruses in the order *Tolivirales*. See Figure 2 legend for details.

**Figure 6 viruses-12-01376-f006:**
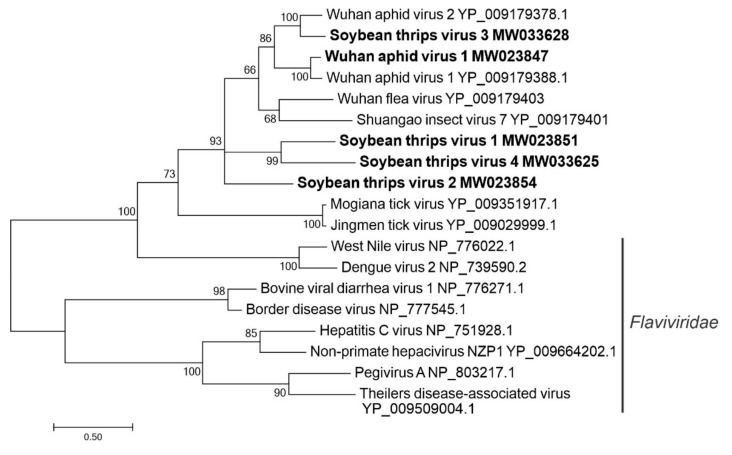
Phylogenetic analysis of the predicted RNA-dependent RNA polymerase amino acid sequences encoded by contigs assembled from soybean thrips RNA-seq data (indicated in bold) and related viruses in the order *Amarillovirales*. See Figure 2 legend for details.

**Figure 7 viruses-12-01376-f007:**
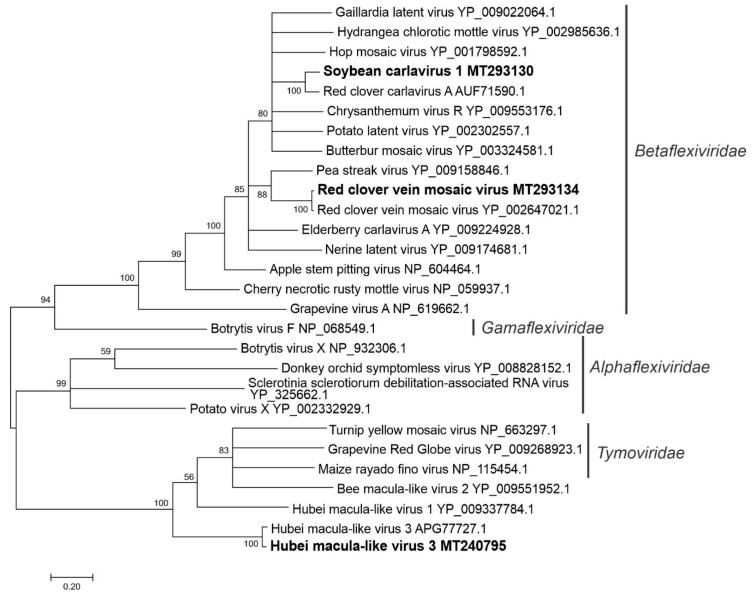
Phylogenetic analysis of the predicted RNA-dependent RNA polymerase amino acid sequences encoded by contigs assembled from soybean thrips RNA-seq data (indicated in bold) and related viruses in the order *Tymovirales*. See Figure 2 legend for details.

**Figure 8 viruses-12-01376-f008:**
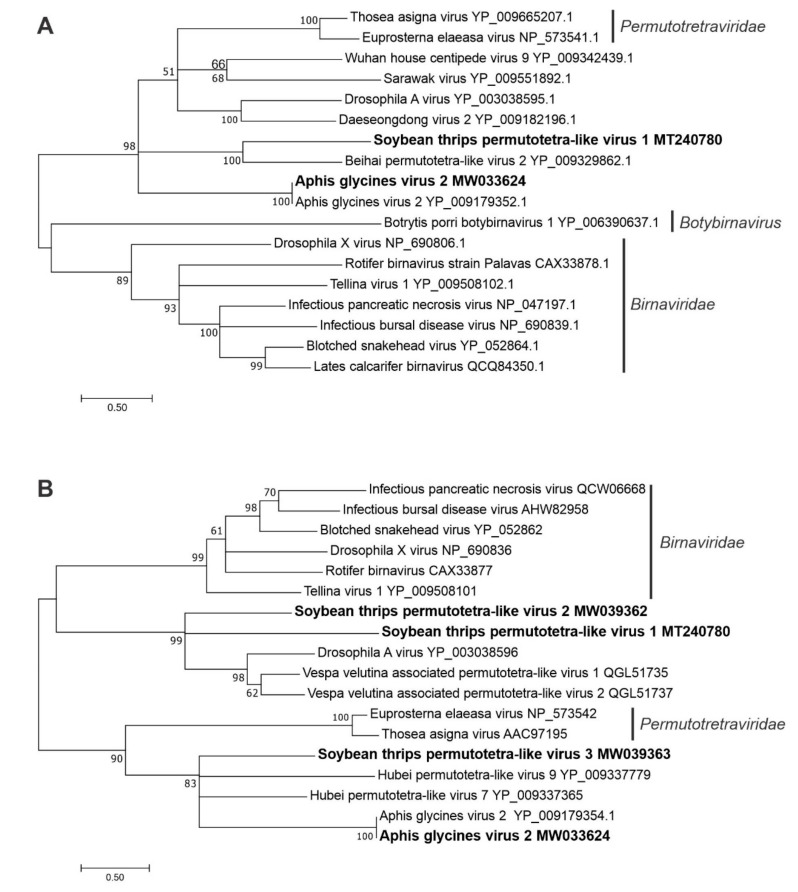
Phylogenetic analysis of the predicted amino acid sequences of RNA-dependent RNA polymerases (**A**) and capsid proteins (**B**) encoded by contigs assembled from soybean thrips RNA-seq data (indicated in bold) and related viruses in the families *Birnaviridae* and *Permutotetraviridae* and genus *Botybirnavirus*. See Figure 2 legend for details.

**Figure 9 viruses-12-01376-f009:**
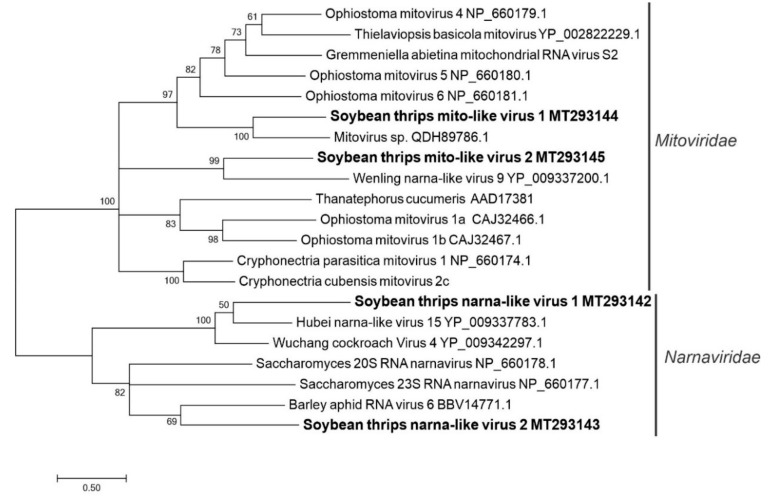
Phylogenetic analysis of the predicted RNA-dependent RNA polymerase amino acid sequences encoded by contigs assembled from soybean thrips RNA-seq data (indicated in bold) and related viruses in the orders *Wolframvirales* and *Cryppavirales*. See Figure 2 legend for details.

**Figure 10 viruses-12-01376-f010:**
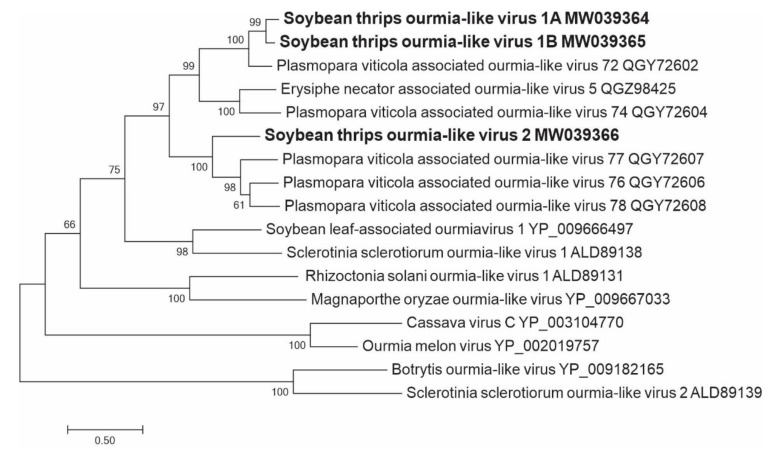
Phylogenetic analysis of the predicted RNA-dependent RNA polymerase amino acid sequences encoded by contigs assembled from soybean thrips RNA-seq data (indicated in bold) and related viruses in the order *Ourlivirales*. See Figure 2 legend for details.

**Figure 11 viruses-12-01376-f011:**
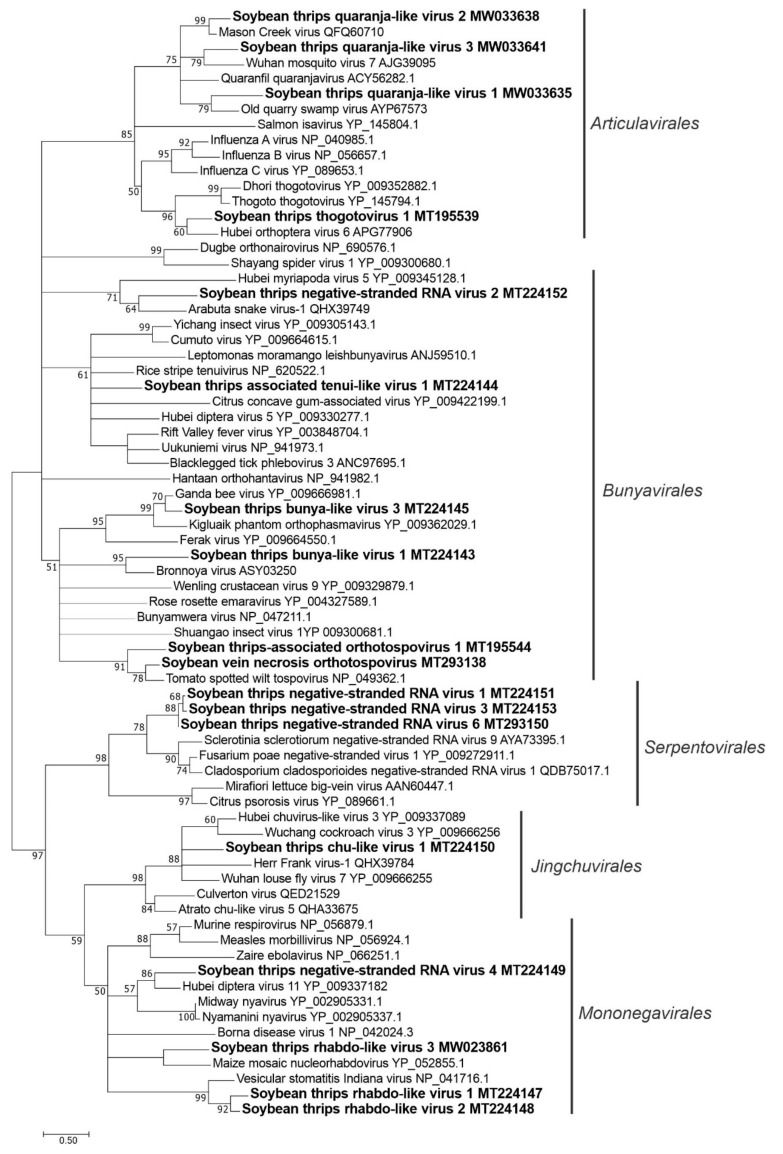
Phylogenetic analysis of the predicted RNA-dependent RNA polymerase amino acid sequences encoded by contigs assembled from soybean thrips RNA-seq (indicated in bold) and related negative-stranded RNA viruses. See Figure 2 legend for details.

**Figure 12 viruses-12-01376-f012:**
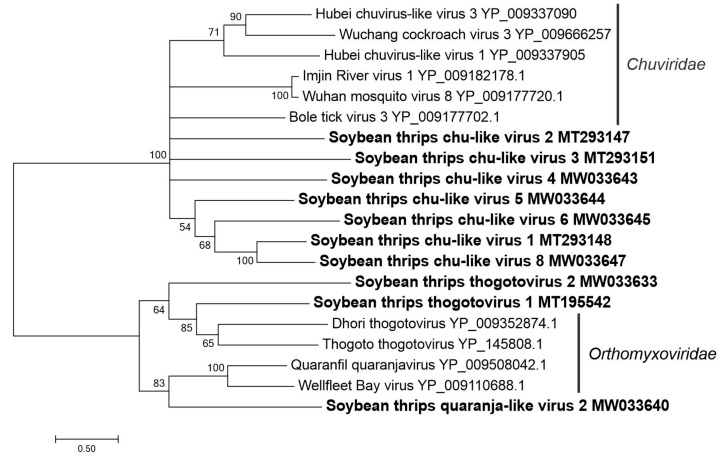
Phylogenetic analysis of predicted glycoprotein amino acid sequences encoded by contigs assembled from soybean thrips RNA-seq (indicated in bold) and related viruses in the order *Jingchuvirales* and *Articulavirales*. See Figure 2 legend for details.

**Figure 13 viruses-12-01376-f013:**
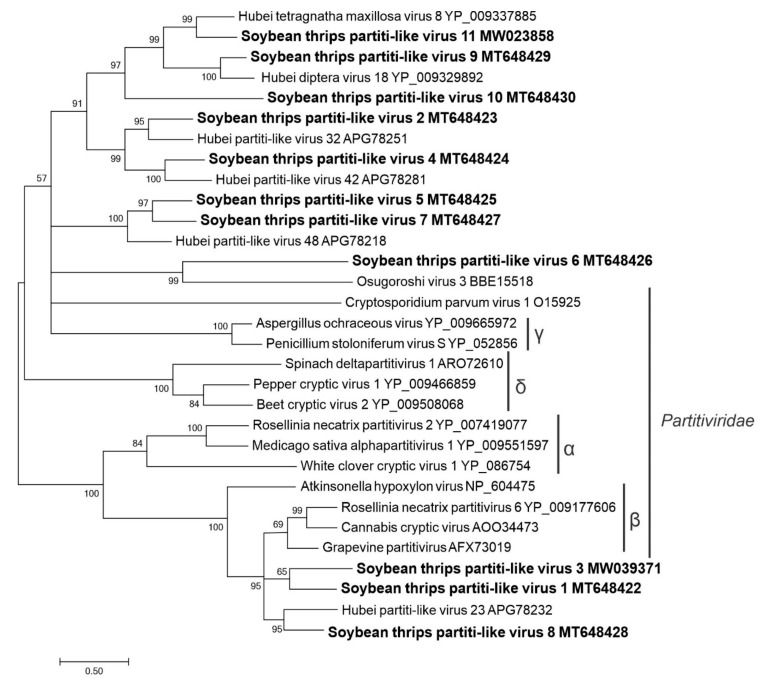
Phylogenetic analysis of the predicted RNA-dependent RNA polymerase amino acid sequences encoded by contigs assembled from soybean thrips RNA-seq data (indicated in bold) and related viruses in the order *Durnavirales*. The genera *Alphapratitivirus*, *Betapratitivirus*, *Deltapartitivirus*, and *Gammapartitivirus* are indicated with Greek letters. See Figure 2 legend for details.

**Figure 14 viruses-12-01376-f014:**
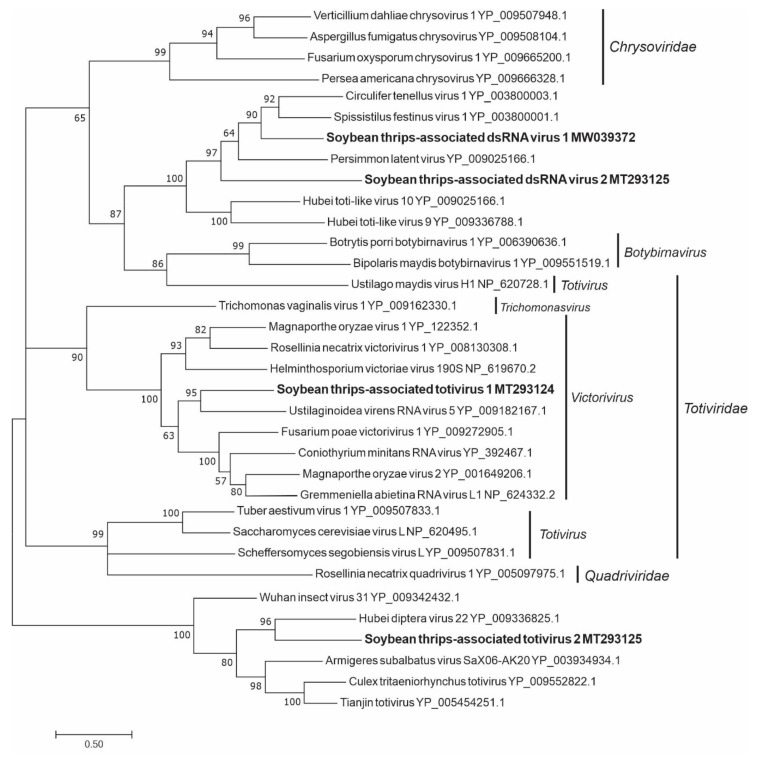
Phylogenetic analysis of the predicted RNA-dependent RNA polymerase amino acid sequences encoded by contigs assembled from soybean thrips RNA-seq data (indicated in bold) and related viruses in the order *Ghabrivirales*. See Figure 2 legend for details.

**Figure 15 viruses-12-01376-f015:**
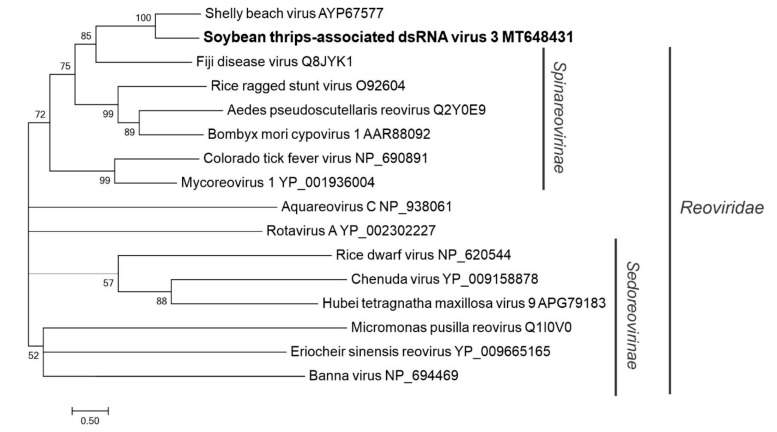
Phylogenetic analysis of the predicted RNA-dependent RNA polymerase amino acid sequences encoded by contigs assembled from soybean thrips RNA-seq data (indicated in bold) and related viruses in the order *Reovirales*. See Figure 2 legend for details.

**Figure 16 viruses-12-01376-f016:**
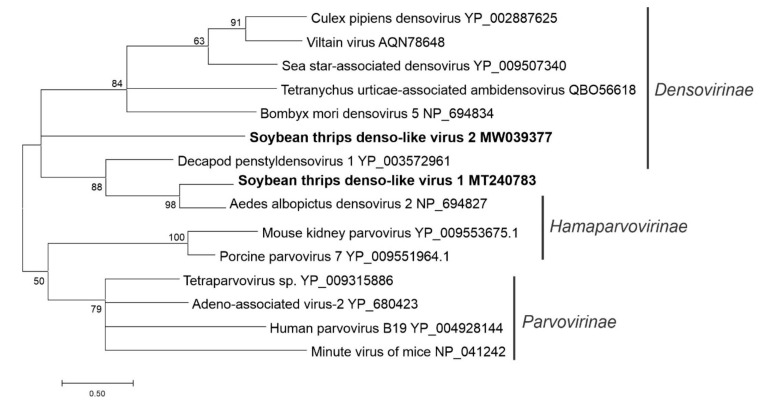
Phylogenetic analysis of predicted NS1 amino acid sequences encoded by contigs assembled from soybean thrips RNA-seq data (indicated in bold)and related viruses in the order *Piccovirales*. See Figure 2 legend for details.

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
