# Peer review of "Soybean Thrips (Thysanoptera: Thripidae) Harbor Highly Diverse Populations of Arthropod, Fungal and Plant Viruses"

_viruses, 2020, doi:10.3390/v12121376_

Round 1

Reviewer 1 Report

This manuscript authored by Thekke-Veetil et al. reported the virus discovery of insect viruses from insect thrips. The writing is straight forward and easy to understand. The major discovery in this report is that thrips contained a lot of RNA viruses that included positive-sense RNA viruses, negative-stranded RNA viruses, and single-stranded DNA viruses. Massive info from this manuscript has been obtained based on RNA-seq. This data set offers tremendous info for further search for viruses in insects. This info can also be used to understand virus interaction in the host, although there were no experimental studies to confirm them.

My major concern for this manuscript is how the authors know these reported RNAs are true viruses, and not from cellular transcripts.  It has been well documented that in cellular DNA such as human DNA, non-retrovirus sequences exist.  In insects, endogenous viral elements (EVE) have been reported that arthropods genomes contain integrated DNA and non-retrovirus (Anneliek M. ter Horst a Jared C. Nigg, Fokke M. Dekker, Bryce W. Falk (2019) Endogenous Viral Elements Are Widespread in Arthropod Genomes and Commonly Give Rise to PIWI-Interacting RNAs JVI.  If the authors could not differentiate cellular viral transcripts and actual viral RNA, then how do the authors claim the reported are indeed actual RNA viruses?  If the authors could not separate these RNAs, the claims for the phylogenetic analysis is not informative since they are not true viruses.  Of course, if the authors can justify that they separated RNA viruses from cellular RNA transcripts, the claims in this manuscript are valid.

Author Response

To address the reviewers comments, text was added that discusses the possibility that some of the reported sequences my be EVEs. We also searched the genomes of three related thrips species for the presence of EVEs, and found a small numbers of EVEs related to glycoprotein genes of negative-strand RNA viruses.  However, the matches were predominantly between repetitive sequences in the host genome and repetitive sequences in the glycoprotein genes.  Hence, the matches may not have been meaningful. We did not find sequences similar to members of the Picornavirales or any viral RdRPs. The analysis is briefly mentioned in the added text.  The section concludes with the statement that, "While endogenous virus-like sequences often are truncated and contain termination codons [81], most of the sequences assembled in this study represent nearly complete virus genomes of positive-sense ssRNA viruses with long ORFs, which suggests that most represent sequences of freely replicating viruses". 

Reviewer 2 Report

This paper presents a detailed bioinformatic analysis of the metatranscriptome of soybean thrips.  As far as I am able to conclude, the study design and data analysis have been carefully executed.  The data presentation is in conventional format but is clear and easy to rad and interpret and thus the conclusions drawn appear to be justified by the data presented.

Whilst this is something of a 'stamp-collecting exercise', nevertheless, it does present for the first time a comprehensive survey of virus genomes or part genomes associated with these important agricultural pests.  It is of interest that the viruses present represent a number of plant and fungal viruses and record a number of new viruses not reported before.  

The paper will be of interest to those involved in pest control programs particularly those involved in surveillance systems.

I didn't detect any typos or changes needed to the text - it was very well written and clear.

Author Response

The reviewer did not suggest changes to the manuscript.

Round 2

Reviewer 1 Report

My questions in the first version of this manuscript was well addressed.  Some transcripts might be insect cellular transcripts that can be difficulty to separated from viral sequences.